# The Wide Range of Antibiotic Resistance and Variability of Genotypic Profiles in *Escherichia coli* from Domestic Animals in Eastern Sicily

**DOI:** 10.3390/antibiotics10010028

**Published:** 2020-12-31

**Authors:** Nunziatina Russo, Alessandro Stamilla, Giuseppe Cascone, Cinzia Lucia Randazzo, Antonino Messina, Massimiliano Lanza, Alessandra Pino, Cinzia Caggia, Francesco Antoci

**Affiliations:** 1Dipartimento di Agricoltura, Alimentazione e Ambiente (Di3A), University of Catania, 95124 Catania, Italy; nunziatinarusso83@gmail.com (N.R.); alessandrostamilla@gmail.com (A.S.); cinzia.randazzo@unict.it (C.L.R.); malanza@unict.it (M.L.); alessandra.pino@unict.it (A.P.); 2Istituto Zooprofilattico Sperimentale of Sicily, 90129 Palermo, Italy; giuseppe.cascone@izssicilia.it (G.C.); antocif@gmail.com (F.A.); 3DVM Consultant Poultry Specialists, via Cava Gucciardo Pirato, 12, 97015 Modica, Italy; vetmessina@gmail.com

**Keywords:** *Escherichia coli*, antimicrobials, multidrug resistance, colistin, PFGE profile, veterinary

## Abstract

The emergence of multidrug resistance among Enterobacteriaceae in livestock poses a serious public health threat. *Escherichia coli*, a usual host of intestinal microbiota, is recognized also as etiological agent of numerous infections widespread in both humans and animals. The colibacillosis is one of the most reported zoonoses worldwide, typically treated with antibiotics in the primary stages. This strategy has promoted the onset of antibiotic-resistant serotypes of *E. coli*, reducing the effectiveness of therapeutic treatments and contributing to antibiotic resistance spread. The current study focused on biodiversity, pathogenicity, and antibiotic resistance profile of 104 *E. coli* strains isolated from domestic animals in Eastern Sicily. The strains were isolated from sick animals and carcasses of six different animal species and screened for resistance against 16 antibiotic molecules, as recommended by WHO and OIE. The antibiotic resistance patterns highlighted that all strains were multi-resistant, showing resistance to at least three antibiotic classes. The highest incidence of resistance was observed against amoxicillin (100%), tylosin (97%), sulfamethoxazole (98%), and erythromycin (92%), while the lowest for colistin (8%). The pathotype characterization identified two EPEC strains and the study of genetic linkage (PFGE) showed a wide variety of profiles. The current study emphasized the wide range of multidrug resistance and genotyping profiles in *E. coli* isolated in Easter Sicily.

## 1. Introduction

*Escherichia coli*, a gram-negative, non-sporulating, facultative anaerobic bacteria, is a natural part of human and warm-blooded animal microbiota, harbored in the colon as a component of the commensal population, living in a symbiotic relationship with the host [1]. Although *E. coli* represents only around 1% of the intestinal microbiota [2], it shows the highest ability to survive, and even grow, outside the host [3,4,5]. This ability is associated with a high phenotypic diversity, mirrored by the high genomic plasticity of *E. coli*, confirmed by the acquisition of numerous mobile genetic elements [6]. Among bacteria, *E. coli* possesses the most diverse lifestyles, and the species includes both commensal and highly pathogenic strains. These latter can cause a wide variety of intestinal and extra-intestinal infections, and some specific serotypes are associated with certain clinical syndromes, and thus serotypes serve as readily identifiable markers that correlate with specific virulent clones [7]. Evolution of *E. coli* is caused by both vertical and horizontal gene transfer. Variable genes, such as plasmids, prophages, and pathogenicity islands (PAIs) [8] make up more than two thirds of the *E. coli* genome. Of major concern is the transmission of virulent and/or resistant *E. coli* between animals and humans through numerous pathways, such as direct contact, contact with animal excretions, or via the food chain. *E. coli* also represents a major reservoir of resistance genes that may be responsible for treatment failures in both human and veterinary medicine. Among animal species, the infection by different *E. coli* strains is widespread and sometimes fatal [9]. One of the most frequently reported diseases worldwide is colibacillosis and, in 2017, Shiga Toxin/Verocytotoxin producing *E. coli* infection has been reported as the fourth most common zoonosis in the EU [10]. Colibacillosis is an extra-intestinal disease characterized by pericarditis, air sacculitis, perihepatitis, peritonitis, that represents a problem of significant economic importance, causing relevant livestock loss [11,12,13]. As known, especially in rural farms, two main pathotypes involved in enteric colibacillosis are the entero-toxigenic *E. coli* (ETEC) and the entero-pathogenic *E. coli* (EPEC) [13,14]. Antimicrobials are the main weapon to fight both the incidence and the mortality associated with colibacillosis [15]. However, typical treatment strategies and, in particular, administration of antibiotics in the primary stages of the disease, have promoted the onset of antibiotic-resistant serotypes of *E. coli*, leading to a reduction in the effectiveness of therapeutic treatments against colibacillosis. In general, the emergence of antibiotic resistance has been documented in *E. coli* isolated from human, animal and environmental sources and represents an emerging health concern [16,17,18]. The selective pressure of antimicrobial use, overuse and misuse in humans and animals comprises the engine driving this process leading to a gradual increase in antibiotic resistance [19]. Consequently, diseases that in the past were treatable are now untreatable or require the latest line of antibiotics [20]. Moreover, the food animal industry contributes to the increasing occurrence of AR through certain farm management practices to promote the well-being and growth of animals that can promote the selection of resistomes in the environment, with potential spillover to animals and humans [21,22,23,24].

Recent surveillance data from the 2000s indicate that, within *E. coli* species, a high antibiotic resistance profile to the major antibiotic classes has been detected [21]. Epidemiological studies in farms are important to provide information about the risk factor for the emergence and persistence of antimicrobial resistance, representing a useful tool to explain the co-selection and the cross-resistance from various antimicrobials and their persistence in the absence of direct selection pressure [25,26]. Indeed, *E. coli* has acquired resistance to cephalosporins, carbapenems, aminoglycosides, fluoroquinolones, and polymyxins. In addition, strains of animal origin often show resistances to other antimicrobial agents, including tetracyclines, phenicol, sulfonamides, trimethoprim, and fosfomycin, and the ability to acquire resistance to colistin, tetracyclines, phenicol, sulfonamides, trimethoprim, and fosfomycin. Molecular characterization of resistant isolates by fingerprinting techniques is an important tool to describe the spread of bacterial clonal units. Among fingerprinting techniques, the pulsed-field gel electrophoresis (PFGE) is particularly useful to demonstrate close relationships among strains, including those manifesting multidrug resistance [18].

The aim of the present study was to evaluate the prevalence of antibiotic resistance of *E. coli* strains isolated from different animal species, presumptively affected by colibacillosis, in order to assess their impact on animal health and food safety. For this purpose, the biodiversity, the pathogenicity and the antibiotic resistance profile of strains isolated from multiple tissue sites of domestic animals were tested.

## 2. Results

### 2.1. Antimicrobial Resistance

Based on EUCAST and CLSI criteria [27,28], all tested strains showed multi-drug resistance against most common veterinary prescribed antibiotics. In detail, in Table 1, the antibiotic resistance profiles of tested strains were reported. Overall, all tested strains showed resistance against at least three antibiotic-classes, resulting in multi-resistance (a comprehensive description of the strain and antibiotic resistance profiles is provided in Appendix A).

The antibiotic resistance patterns highlighted that three out 104 (2.9%) strains showed resistance against 15 among the 16 tested antibiotics. The most of tested strains, 17 out 104 (16.3%), showed resistance to nine antibiotics, 11 strains (10.6%) showed resistance to 10 molecules, 10 strains (9.6%) against 12 molecules and other 10 strains against eight molecules. Zooming on the resistance distribution, the highest incidence was observed against amoxicillin (100%), tylosin (97%), sulphamethoxazole (94%), and erythromycin (92%), whereas lower incidences were observed for aminoglycosidic antibiotics, with the 29% and 32% of strains resistant against aminosidine and apramycin, respectively (Table 1). 

To investigate the differences of *E. coli* strains isolated from bovine and poultry animals, based on antibiotic resistance profiles, principal component analysis (PCA) was employed. Score plot is effective in showing the difference among resistance profile of *E. coli* strains and in separating them in the graph. Overall, three main groups (A, B and C) were detected (Figure 1). In detail, closer distances were detected among both bovine and poultry isolates, clustered into the group A, that includes strains with the highest level of resistance. On the contrary, into the group B, clustering strains with higher level of susceptibility, a great dispersion for strains from bovine was observed. Although less closely spaced, the strains EC92, EC105, and EC 113, that exhibited susceptibility to aminosidine, colistin, lincomycin and spectomicin, tylmicosin, trimethoprim and apramycin, and categorized as intermediate for sulphamethoxazole, were grouped into group C. Finally, the strains EC114 and EC103, positioned far away from other groups, were distinguished to show resistance to antibiotics different to those listed in the opposite quarter (Figure 1).

Although the lowest incidence (8%) of resistance was observed to colistin, the eight resistant strains showed a MIC value of 16 µg/mL (data not shown), which is much higher than the clinical breakpoint [23]. Furthermore, the colistin-resistant strains, despite isolated from different domestic animals, were distinguished for the high level of resistance, spanning from 15 to 8 antibiotics (Table 2). Focusing on the single antibiotic patterns, all the eight colistin-resistant *E. coli* strains (EC11, EC23, EC33, EC36, EC59, EC62, EC83, EC85) showed concurrent resistance to tylosin, sulphamethoxazole, and amoxicillin. Among them, the EC23 strain showed sensibility to aminosidine, tylmicosin, and apramycin and the EC33 distinguished from others for the highest level of susceptibility and the highest level of intermediated categorization to thiamphenicol, ampicillin, doxycycline, and flumequine (Table 2). The antibiotic resistance profiles observed for the two EPEC strains showed a high degree of variability, with the EC15 strain exhibiting resistance to only three antibiotics (tylosin, sulphamethoxazole and amoxicillin) and the EC15 strain exhibiting susceptibility exclusively to colistin as well as lincomycin and spectomicin antibiotics and resistance against all other 14 tested antibiotics (Table 2).

Interestingly, 42 out 104 strains (40%) were grouped into the intermediate category for resistance to thiamphenicol, which could be interpreted as a launch into resistance acquisition. 

### 2.2. DEC Phatotypes Detection

The multiplex PCR assay, applied in the present work for the rapid detection of specific categories of pathogenic types, revealed that none of the tested strains showed the tested genes for EAEC, STEC, ETEC, and EIEC pathotypes. Only two strains, the EC15 and the EC31, isolated from dog breast and broiler liver, respectively, were found harboring the *eae* gene (EPEC pathotype). 

### 2.3. PFGE Analysis

PFGE was performed in order to study the genetic linkage among *E. coli* strains isolated from different domestic animals. Data showed that the *Xba*I digestion allowed obtaining a range, spanning from 7 to 21, visibly distinguishable fragments (Figure 2). Indeed, in the present study, the technique allowed to cluster 70 out of 104 of *E. coli* tested strains. The analysis of dendrogram showed a wide variety of profiles, with any correlation between a specific profile and a defined animal species, isolation site, or year of isolation. As previously reported, these data suggest a great variety among *E. coli* strains shed in the domestic animals setting and into the environment in general [18,29].

## 3. Discussion

It is well known that *E. coli*, although a normal inhabitant of the intestinal tract, can also be associated with a variety of pathological conditions in both humans and farm animals [30]. Moreover, the largest resistance to numerous antimicrobial agents of interest of both human and veterinary fields observed in *E. coli* strains represents a serious problem to global public health, resulting in a significant impact on animal health and food safety [31]. Colibacillosis, an extra-intestinal disease caused by ETEC and EPEC *E. coli* pathotypes, represents one of the most frequently reported disease worldwide in the livestock farm, commonly countered by antibiotics [11,12,13] and thus a further worrying source of spread of antibiotic resistance.

The present investigation was conducted to achieve phenotypic resistance profile against the commonly veterinary prescribed antibiotics in clinical strains, isolated from different domestic animals, probably affected by colibacillosis. Results of DEC detection, through multiplex PCR analysis, showed that the majority of the strains did not show the presence of the intimin adhesin encoded by the *eae* gene from the LEE (locus of enterocyte effacement) pathogenicity island, with the exception of two strains. The implications of the presence of pathogenic and/or commensal strains in the intestinal environment are not clearly discernible, although it can be speculated that this could be a leading source of a diverse range of EPEC strains, as reported for atypical enterohaemorrhagic and enteropathogenic *E. coli* [32,33]. However, the diversity within both *E. coli* sero-groups and sero-types is commonly reported [34], confirming the heterogeneous nature of EPEC in terms of virulence features. Indeed, well-recognized virulence factors do not occur universally among EPEC, suggesting the presence of multiple alternative mechanisms mediating pathogenicity. Finally, in addition to the virulence status of the strain, host status and presence and type of predisposing factors must be considered for their influence in the infection symptoms and severity [32].

In the present study, all strains showed multi-resistance, being resistant against at least three antibiotic classes. Our findings are in agreement with previous reports highlighting the emergence, propagation, accumulation, and maintenance of antimicrobial-resistant pathogenic *E. coli* in human and veterinary medicine [35,36,37,38]. The phenotypic antibiotic resistance patterns showed high resistance to the most common veterinary antibiotics such as amoxicillin, tylosin, sulfamethoxazole, and erythromycin. The high degree of observed resistance could be explained by the fact that many antimicrobials are used to treat domestic animal infections, and in the past, as growth promoters or preventative measures [18,39,40]. Such a high level of resistance to amoxicillin is probably related to the popularity of the β-lactams, in both human and animal disease treatments [39]. Moreover, in *E. coli* species, one of the most remarkable phenomena is the rapid increase of plasmid-mediated beta-lactam resistance, which contributes to a faster diffusion of resistance in different environments [41,42]. The routine exposure, for extended periods, to sub-therapeutic doses of antimicrobial agents by livestock species contributes to a significantly higher prevalence of resistance, compared to species that are typically exposed to therapeutic doses for brief periods [39,43]. Likewise, the approved use of tylosin in feed, as a preventive measure for liver abscess, has probably contributed to the relative resistance increase [41,42].

Regarding the high incidence of erythromycin- and sulphamethoxazole-resistance, the results of the present study are in agreement with those recently reported [44] on *E. coli* isolated from different sources. Although several reports on *E. coli* highlighted a wide distribution of colistin resistance, frequently mediated by the mobile *mcr* gene [44], in the present study, the percentage of resistance against colistin was low. This finding is great of interest because colistin is recognized as the last-resort antibiotic for the treatment of infectious diseases caused by multidrug-resistant (MDR) gram-negative bacteria [44]. Moreover, as recommended by the EUCAST subcommittee [45], in the present study the resistance to colistin was confirmed by determination of MIC through broth micro-dilution method. According to the EUCAST [46] and Loose et al. [47], a strain is defined as colistin-resistant (for *mcr-1* gene presence) when the MIC value ranged between 4.0–16 mg/L and colistin-susceptible (commonly for the non-*mcr-1* presence) when the MIC values ranged between 0.25–1.0 mg/L. Hence, the MIC value of 16 mg/L, revealed in the present study, can be attributed to the presence of plasmid-transferred mobilized colistin resistance *mcr-1* gene [48,49,50]. Moreover, the colistin resistance was always related to resistance to tylosin, sulphamethoxazole, and amoxicillin, as recently reported [44].

Furthermore, the intermediated categorization obtained for thiamphenicol antibiotics (chloramphenicol representative derivative), synonym of observed reduced susceptibility, suggests the possibility of acquiring resistance [51]. Amphenicols present a broad-spectrum inhibitory effect on both gram-positive and -negative bacteria but are particularly effective against the latter. The severe levels of resistance to amphiphenicles may be attributable to spread or abuse in the animal industry, which may have increased selective pressure on *E. coli* [52]. The antibiotic resistance profiles observed for the two EPEC strains showed high variability, confirming the importance of the considerable polymorphism of genes encoding for this virulence factor. Likewise, the acquisition and the exchange of the virulence gene among pathogenic *E. coli* strains are believed to provide an evolutionary pathway to pathogenicity [53]. Moreover, no correlation was observed between phenotypic antibiotic resistance and reference animal and/or isolation site, or year of isolation.

Finally, the clonal relationship among strains by *Xba*I-PFGE was explored. Macro-restriction analysis by PFGE is capable of clustering and differentiating many pathogens due to its sensitivity and discriminatory power. Although both traits might be affected by the organism and the used restriction enzyme, PFGE has a high epidemiologic relevance and is the “gold standard” method for subtyping bacterial pathogens [54]. Typing methods for discriminating different bacterial isolates of the same species are an essential epidemiological tool in infection prevention and control. However, although PFGE is considered as the “gold standard”, many strains are not typable by this technique due to the degradation of the DNA during the process (gel smears) [29,33]. In the current study, the observed high genetic diversity, highlighted by the unique pattern pulso-types, confirms the independent origin of the strains, being genetically distinct from one another. This fact might be explained by the large genomic diversity of *E. coli* isolates, also confirmed by applying multi-locus enzyme electrophoresis technique [55,56]. Our results showed no correlation with any PFGE patterns within the same animal sample, tissue site or among the same geographical location, or on the year of isolation. These results were predictable because, as previously reported, the different isolation sources involved genetic differences of strains [57]. Moreover, no correlation was observed between the antibiotic resistance patterns and PFGE types. The possibility of horizontal transfer of mobile genetic elements to confer resistance, such as plasmids, integrons, or phage-mediated exchange, could likely contribute to different PFGE patterns, as observed here [58].

The present study, offering a snapshot of antibiotic the resistance profile of *E. coli* strains commonly isolated from livestock farms, underscores the need for a clear assessment of drug resistance of *E. coli* as a normal inhabitant of the intestinal tract. Indeed, numerically dominant commensals and not frank pathogens, represent the dominant lineages here observed. Further studies (e.g., surveillance) are indicated to identify the role of nonpathogenic *E. coli* strains and risk factors associated with antibiotic resistance spread.

## 4. Materials and Methods

### 4.1. Source of E. coli Isolation

In the present work, a total of 104 *E. coli* strains were tested. In detail, the strains were previously isolated and identified at the Istituto Zooprofilattico Sperimentale of Sicily, in Ragusa, from June 2015 to December 2019. The strains were isolated from necroscopies of various domestic animals, (*n* = 68) poultry, (*n* = 24) bovine, (*n* = 7) swine, (*n* = 1) buffalo, (*n* = 1) canine, (*n* = 1) canary, (*n* = 1) equine, (*n* = 1) rabbit that died due to colibacillosis or showed symptoms connected with *E. coli* infection or the onset of concomitant diseases. In detail, a total of 102 swabs were analyzed, 96 from of organs and six from breast swabs (from canine, equine, and bovine), one from bovine cephalorachidian fluid and two from bovine and equine crusts. The organs were aseptically withdrawn, placed in cabin, burned in the surface, and cut longitudinally with scalpel approximately 3 cm. The internal fluid samples were picked up via swab inserted inside the trimmed area. The swab of each sample was streaked onto a MacConkey’s agar plate and incubated a 37 °C for 24 h. Suspected *E. coli* colonies were re-plated onto blood agar in duplicate. The colonies were identified using two different methods: The API^®^20E (Biomerieux, Nouveautės, France) and the GN card with VITEK^®^2 (Biomerieux, France) tests. Analyses were performed according to the manufacturer’s instructions, using the specific software: ApiwebTM and VITEK 2 Systems, version 08.02. The *E. coli* isolates were stored at −80 °C in CRYOBANK^®^ (Mast Group, Bootle, UK) until the use.

### 4.2. Susceptibility Test

The 104 *E. coli* strains were tested for antibiotic susceptibility against 16 antimicrobial agents, using disk agar diffusion method, according to the Clinical Laboratory Standard Institute (CLSI) [28] and EUCAST guidelines [27]. Antibiotics were selected based on the recommendations of the World Health Organization (WHO) and World Organization for Animal Health. The tested antibiotics belonged to 11 classes of antimicrobial agents were: Aminosidine (AMI, 60 µg), Colistin (COL, 10 µg), Enrofloxacin (ENR, 5 µg), Lincomycin/Spectinomycin (LIN/SPE 2 + 100 µg), Oxytetracycline (OXY, 30 µg), Thiamphenicol (THP, 30 µg), Tilmicosin (TLM, 15 µg), Tylosin (TYL, 30 µg), Trimethoprim (TRM, 1.5 µg), Sulphamethoxazole (SUL, 25 µg), Ampicillin (AMP, 10 µg), Doxycycline (DOX, 30 µg), Flumequine (FLU, 30 µg), Erythromycin (ERY, 15 µg), Amoxicillin (AMX, 25 µg) and Apramycin (APR, 30 µg). For the colistin-resistant strains, the minimum inhibition concentrations (MIC) were determined by broth microdilution method, using the MIC-Strip Colistin (MERLIN Diagnostika GmbH, Bornheim, Germany), according to the manufacturer’s instructions, following the International Standard reference method (ISO 20776-1) [59], as recommended by the EUCAST subcommittee [45].

### 4.3. DEC Pathotypes Investigation

The pathogenicity of individual *E. coli* lineages is mostly related to virulence gene content [29]. For this reason, the screening for specific virulence genes, defining the five most relevant DEC pathotypes, was performed in the present study. As reported by Ori et al. (2018) [34], a multiplex PCR assay was carried out for the EPEC, EAEC, STEC, ETEC and EIEC pathotypes. In detail, the following genes were detected: For EPEC, the *eae*, responsible for the production of the adhesin intimin; for EAEC, the *aat*A, encoding for a protein related to an ATP-binding cassette transport system; for STEC, the *stx*1 and *stx*2, related to the production of the *stx*1 and *stx*2 toxins respectively; for ETEC, the *lt*A and *st*A, related to LT and ST toxins production; for EIEC, the *ipa*H, associated with the invasion plasmid antigen H [34]. Primer sequences and amplicon sizes are described in Table 3. Template DNA for PCR reactions was produced by dissolving *E. coli* colony, cultivated on Tryptic Soy agar, in 20 µL of DNAasi free water. The PCR mixture was performed in a final volume of 25 μL, consisting of 24 μL of DreamTaq master mix 2X (Thermo Fisher Scientific, Rodano, Italy), 0.5 μL of each genus primer, and 1.0 μL of DNA template, previously obtained. The thermo cycle conditions were: 40 cycles of 95 °C 5 min, 95 °C 40 s, 58 °C 1 min, 72 °C 2 min. PCR products were electrophoresed in 1.5% agarose gels gel containing Gel Red Nucleic Acid stain (Biotium, Fremont, USA) in 0.5 TBE (25 mM Tris-borate, 0.5 mM EDTA), and photographed under UV light (Axygen, Gel Documentation System). Amplicon sizes from each DEC sample were compared to those in the control strains to compose a genotype for each representative DEC group member.

### 4.4. Pulsed-Field Gel Electrophoresis (PFGE)

The macro-restriction fragment separation by PFGE was performed using the 24-h PulseNet standardized PFGE protocol for *E. coli* non-O157:H7 [60]. Digestion was carried out with 50 U of *Xba*I restriction enzyme (Thermo Fisher Scientific, Rodano, Italy) for 2 h at 37 °C, after pre-digestion in buffer for 5–10 min at 37 °C. DNA fragments were resolved in 1% agarose gel in 0.5X TBE electrophoresis buffer at 14 °C, using the CHEF-DR III system (Bio-Rad Laboratories, Hercules, CA, USA). The runtime was 20 h, with a constant voltage of 6 V/cm, using a linear pulse ramp of 6.76–35.58 s. PFGE images of gels were captured using Gel Documentation System (Axygen Scientific, Torino, Italy).

### 4.5. Data Analyses

The TIFF images obtained by PFGE were analyzed with a temporary BioNumerics evaluation license from Applied Maths (version 8.0 software package, Applied Maths, Sint-Martens-Latem, Belgium) from which we received permission to publish. The relatedness among the patterns was estimated by the proportions of shared bands, after applying the Dice coefficient. The UPGMA method was used to generate dendrograms with 1.5% tolerance values. The analysis of the patterns was confirmed visually.

Data obtained from antibiotic resistance between *E. coli* strains isolated from bovine and poultry animals were subjected to principal component analysis (PCA), using XLSTAT (2016), in order to achieve higher data compression efficiency.

## 5. Conclusions

In the present study, the high incidence of observed resistance against the most common antibiotics used in both human and veterinary practices, poses concern about the antibiotic resistance within the *E. coli* species. The proportion and the diversity of multidrug-resistant phenotypes among *E. coli* isolates, even within the same pathogenic lineage, confirm a high probability of horizontal gene transfer. However, the usefulness of colistin, a last-resort antibiotic for the treatment of resistant gram-negative bacterial infections, was not compromised in this study. The observed high genetic diversity, highlighted by the unique pattern pulso-types obtained, confirms the independent origin of the strains, and the large genomic diversity within *E. coli* species. Further studies for DEC strains characterization are required in order to in-depth investigate the heterogeneous nature of EPEC strains. Furthermore, continuous surveillance to understand the role of non-pathogenic transmissible clones is desirable in order to recognize new patterns of virulence and identify the risk factors associated with the antibiotic resistance spread

## Figures and Tables

**Figure 1 antibiotics-10-00028-f001:**
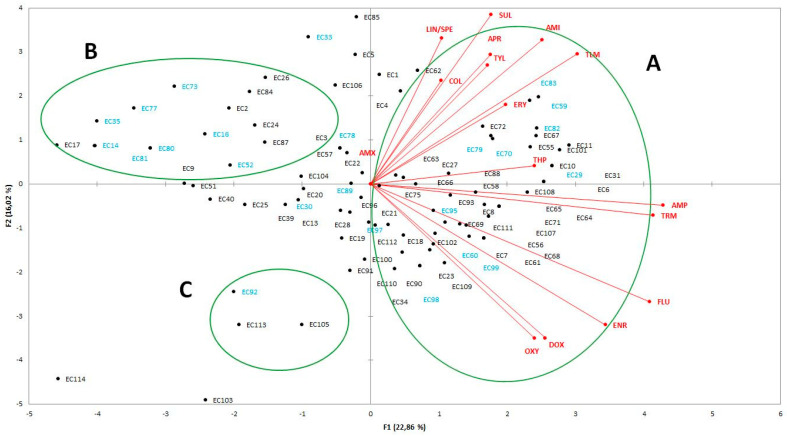
Biplot of principal component analysis (PCA) analysis showing the antibiotic resistance distribution among *E. coli* strains isolated from bovine (in blue) and poultry (in black).

**Figure 2 antibiotics-10-00028-f002:**
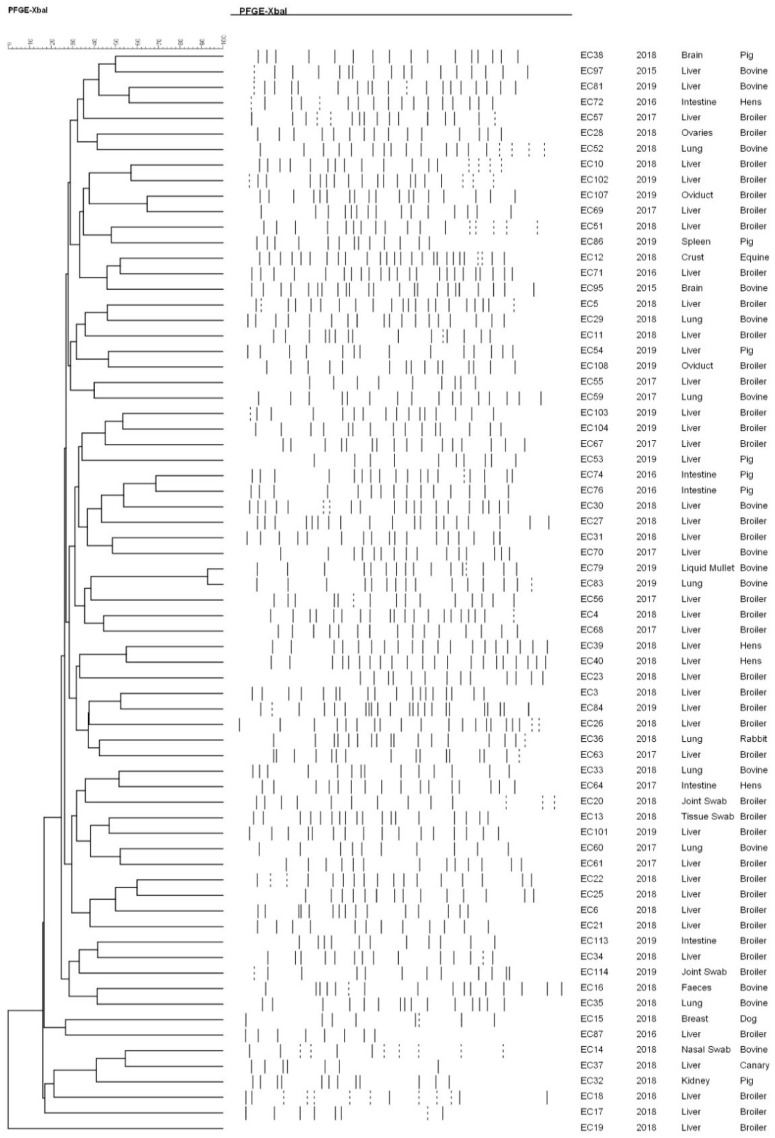
*Escherichia coli* phylogenetic three based on pulsed-field gel electrophoresis (PFGE) analysis.

**Table 1 antibiotics-10-00028-t001:** Antibiotic-resistance pattern of *E. coli* population.

Antimicrobial Agents	No. Tested	Resistant		Intermedium		Susceptible	
		no.	%	no.	%	no.	%
Aminosidine	104	30	29	1	1	73	70
Colistin ^1^	104	8	8	0	0	96	92
Enrofloxacin	104	48	46	10	10	46	44
Lincomycin and Spectomicin	104	34	33	22	21	48	46
Oxytetracycline	104	71	68	20	19	13	13
Thiamphenicol	104	52	50	42	40	10	10
Tylmicosin	104	52	50	1	1	51	49
Tylosin	104	101	97	1	1	2	2
Trimethoprim	104	58	56	2	2	44	42
Sulphamethoxazole	104	98	94	6	6	0	0
Ampicillin	104	78	75	8	8	18	17
Doxycycline	104	68	65	18	17	18	17
Flumequine	104	56	54	8	8	40	38
Erythromycin	104	96	92	7	7	1	1
Amoxicillin	104	104	100	0	0	0	0
Apramycin	104	33	32	1	1	70	67

^1^ Minimum Inhibitory Concentration (MIC) was determined by the micro-broth dilution method using the MIC-strip kit (MERLIN Diagnostika GmbH, Bornheim, Germany).

**Table 2 antibiotics-10-00028-t002:** Antibiotic resistance pattern of clostin-resistant and EPEC^+^
*E. coli* strains.

Antimicrobial Agents	Colistin-Resistant Strains								EPEC^+^ Strains	
	EC11	EC23	EC33	EC36	EC59	EC62	EC83	EC85	EC15	EC31
Aminosidine	R	S	R	R	R	R	R	R	S	R
Colistin	R	R	R	R	R	R	R	R	S	S
Enrofloxacin	R	R	S	R	S	S	R	S	S	R
Lincomycin and Spectomicin	S	S	S	R	I	I	R	R	S	S
Oxytetracycline	R	R	S	R	R	R	R	I	S	R
Thiamphenicol	R	R	I	I	R	R	I	I	S	R
Tylmicosin	R	S	R	I	R	R	R	R	S	R
Tylosin	R	R	R	R	R	R	R	R	R	R
Trimethoprim	R	R	S	R	R	S	R	S	S	R
Sulphamethoxazole	R	R	R	R	R	R	R	R	R	R
Ampicillin	R	R	I	R	R	R	R	R	S	R
Doxycycline	R	R	I	R	R	R	I	I	I	R
Flumequine	R	R	I	R	R	S	R	S	S	R
Erythromycin	R	I	R	R	R	R	R	R	I	R
Amoxicillin	R	R	R	R	R	R	R	R	R	R
Apramycin	R	S	R	R	R	R	R	R	S	R

R: Resistant; I: Intermedium; S: Susceptible.

**Table 3 antibiotics-10-00028-t003:** Virulence factor primer sequences and amplicon sizes used in this study.

Target Genes	Primer Sequences (5′-3′)	Amplicons’ Size (bp)
*stx1*	ATAAATCGCCATTCGTTGACTAC AGAACGCCCACTGAGATCATC	188
*stx2*	GGCACTGTCTGAAACTGCTCC TCGCCAGTTATCTGACATTCTG	255
*eae*	GACCCGGCACAAGCATAAGC CCACCTGCAGCAACAAGAGG	384
*aatA*	CTGGCGAAAGACTGTATCAT CAATGTATAGAAATCCGCTGTT	630
*ipaH*	CTCGGCACGTTTTAATAGTCTGG GTGGAGAGCTGAAGTTTCTCTGC	917
*ltA*	GGCGACAGATTATACCGTGC CGGTCTCTATATTCCCTGTT	450
*stA*	ATTTTTMTTTCTGTATTRTCTT CACCCGGTACARGCAGGATT	190

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
