# Peer review of "The Wide Range of Antibiotic Resistance and Variability of Genotypic Profiles in Escherichia coli from Domestic Animals in Eastern Sicily"

_antibiotics, 2020, doi:10.3390/antibiotics10010028_

Round 1

Reviewer 1 Report

In the present study, the authors, Russo et al. Conducted a study on the antibiotic resistance profile of E. coli strains, isolated from livestock farms. The antibiotic resistance patterns revealed that strains were multi-resistant to at least 3 antibiotic classes (amoxicillin, tylosin, sulfamethoxazole and erythromycin).The authors explained the probable mechanism for this antibiotic resistance and conducted the necessary experiments for this study.  To assess the impact of usage of antibiotics in animal health and food safety this study known to be interesting. 

In this content, I recommend this article for publication in the journal Antibiotics.    After minor revision. Comments:

  1. Line 39 [0]? 
  2. Introduction section includes recent study 
  3. Since study relates to antibiotic resistanceintroduction about the harmful effect of antibiotic resistance in animal industry to be mentioned

Author Response

Dear referee, thank you very much for your suggestions. 

We enriched the introduction section adding different scientific evidences regarding the effect of antibiotic resistance in animal industry (lines 68-72)

Reviewer 2 Report

General comments

The problem of antimicrobial resistance is very important and the availability of studies on the spread of these problems in animal population is very interesting.

This paper reports the data on several year and several pathologies where apparently E.coli were isolated. However, the results showed as nearly all the isolated were not classified within intestinal pathogenic E.coli, despite at page 2 line 83 it is stated “presumptively affected by colibacillosis” Therefore, it is necessary to increase the information on the source of these isolates (i.e. they are the only bacteria species isolated, if the final diagnosis was “colibacillosis” …). If possible, it would be useful to analyse these isolates to identify if they can be classified as ExPEC or as other pathotypes.

Specific comments

Page 7: discussion lines 160-170. This paragraph tries to justify the absence of EPEC and ETEC isolates, since these was the “selection” criteria for the study. A relatively poor retrieval rate for these pathotype is not unexpected, but the explanation supplied is not acceptable since several isolates are not from gut and there are other potential explanations for this outcome. This aspect must be discussed with more details and with a broaden approach.

page 8 line253 : why macrolides, which are not effective against E.coli have been included in the sensitivity assays? Please supply information on the rationale. These aspects must be considered also in discussion (i.e. page 7 line 175 where high resistance is attributed to the wide use of tylosin and not to the intrinsic resistance).

Further comments

There are several typewriting or format mistakes, please carefully revise the manuscript before submitting it.

Page 1 lines 74-78:  one or more references are needed to support statement

Page 3 table 1: there are format problems

Author Response

Dear referee,

thank you for your comment. In the Rev paper  the source of E. coli isolates (lines 259-261) was better explained.

We clarified that E. coli strains originated from necroscopies of animal’s organs died either from colibacillosis or for the onset of concomitant diseases (lines 261-262).

Specific comments

Page 7: discussion lines 160-170. This paragraph tries to justify the absence of EPEC and ETEC isolates, since these was the “selection” criteria for the study. A relatively poor retrieval rate for these pathotype is not unexpected, but the explanation supplied is not acceptable since several isolates are not from gut and there are other potential explanations for this outcome. This aspect must be discussed with more details and with a broaden approach.

R:Thank you for your indication. We improved this aspect (lines 186-189)

page 8 line253 : why macrolides, which are not effective against E.coli have been included in the sensitivity assays? Please supply information on the rationale. These aspects must be considered also in discussion (i.e. page 7 line 175 where high resistance is attributed to the wide use of tylosin and not to the intrinsic resistance).

R: Thank you for your comment. Although macrolide antibiotics explain no effect against E. coli, some molecules are part of the antibiotics panel used in routine laboratory diagnostics. For this reason, macrolides were included in the present study. Mass medication is the most feasible means of treatment in intensive farming, but with the possibility of drug dispersal into the environment as well as the related emergence of antibiotic-resistant bacteria following continuous exposure to sub-therapeutic doses. Consequently, we believe that the prolonged tylosin exposure, recently approved by the U.S. Food and Drug Administration for continuous inclusion in feed for liver abscess prevention and in the past widely used in Europe as growth promoter, not only could select to E. coli strains with higher levels of tylosin resistance, but also could contribute to the selection for strains resistant to other macrolides.

Levy, S.B.; Marshall, B. Antibacterial resistance worldwide: Causes, challenges and response. Nature Medicine. Nature Medicine Supplement 2004, 10.

Meseko, C.; Makanju, O.; Ehizibolo, D.; Muraina, I. Veterinary Pharmaceuticals and Antimicrobial Resistance in Developing Countries. Veterinary Medicine and Pharmaceuticals, 2019.

Indranil, S.; Samiran, B. The emergence of antimicrobial-resistant bacteria in livestock, poultry and agriculture. Antimicrobial Resistance in Agriculture. Perspective, Policy and Mitigation, 2020 19-27.

Further comments

There are several typewriting or format mistakes, please carefully revise the manuscript before submitting it.

R: Thank you. We corrected them.

Page 1 lines 74-78:  one or more references are needed to support statement

R: Thank you for your suggestion. We improved the scientific evidences.

Page 3 table 1: there are format problems

R: Sorry for the mistake. We corrected it.

Round 2

Reviewer 2 Report

The Authors' answers to my comment are satisfactory and now the paper is acceptable for publication.

Author Response

Please find the revised version of the paper in the attachment.
